# MMAL: Multi-Modal Analytic Learning for Exemplar-Free Audio-Visual Class Incremental Tasks

## ABSTRACT

Class-incremental learning poses a significant challenge under an exemplar-free constraint, leading to catastrophic forgetting and sub-par incremental accuracy. Previous attempts have focused primarily on single-modality tasks, such as image classification or audio event classification. However, in the context of Audio-Visual Class-Incremental Learning (AVCIL), the effective integration and utilization of heterogeneous modalities, with their complementary and enhancing characteristics, remains largely unexplored. To bridge this gap, we propose the Multi-Modal Analytic Learning (MMAL) framework, an exemplar-free solution for AVCIL that employs a closed-form, linear approach. To be specific, MMAL introduces a modality fusion module that re-formulates the AVCIL problem through a Recursive Least-Square (RLS) perspective. Complementing this, a Modality-Specific Knowledge Compensation (MSKC) module is designed to further alleviate the under-fitting limitation intrinsic to analytic learning by harnessing individual knowledge from audio and visual modality in tandem. Comprehensive experimental comparisons with existing methods show that our proposed MMAL demonstrates superior performance with the accuracy of 76.71%, 78.98% and 76.19% on AVE, Kinetics-Sounds and VGGSounds100 datasets, respectively, setting new state-of-the-art AVCIL performance. Notably, compared to those memory-based methods, our MMAL, being an exemplar-free approach, provides good data privacy and can better leverage multi-modal information for improved incremental accuracy.

## CCS CONCEPTS

• **Computing methodologies → Artificial intelligence**; **Computer vision**; **Computer vision tasks**; **Scene understanding**.

## KEYWORDS

multi-modal, incremental learning, analytic learning

**ACM Reference Format:**
Anonymous Author(s). 2024. MMAL: Multi-Modal Analytic Learning for Exemplar-Free Audio-Visual Class Incremental Tasks. In *Proceedings of the 31st ACM International Conference on Multimedia (MM '24), October 28–November 1, 2024, Melbourne, MEL, Australia.* ACM, New York, NY, USA, 10 pages. https://doi.org/10.1145/nnnnnnn.nnnnnnn

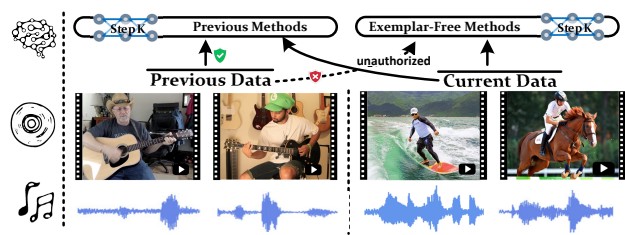

**Figure 1: Illustration of the audio-visual class-incremental learning (AVCIL) task w/ and w/o exemplar.**

## 1 INTRODUCTION

The inherent correlation between audio and visual signals empowers humans to associate sounds with their sources, such as identifying a barking dog. By leveraging the enhancing and complementary nature of audio and vision, these endeavors aim to enhance the understanding and capabilities of models for tasks such as speech recognition [1, 8, 37, 52, 61], sound localization [12, 14, 22, 23, 31, 39, 47, 48, 50] and audio-visual event classification [7, 16, 55, 60, 65]. The aforementioned studies demonstrated the effectiveness of jointly modeling audio-visual modalities in capturing meaningful cross-modal semantic correlations. Inspired by their success, in this paper, we tackle the problem of identifying the sounding objects in synchronized audio-video streams. To be mentioned, it is crucial for real-world applications where the set of classes (e.g., types of sounds or visual events) may evolve or where the model needs to integrate new information into an ever-growing knowledge base. By focusing on preserving semantic similarity and learned correlations between modalities, we aim to mitigate catastrophic forgetting (i.e., a model loses previously acquired knowledge or experiences a significant performance decline when learning new information) and enhance the resilience and adaptability of audio-visual systems.

Class-incremental learning (CIL) [49, 62] progressively updates network parameters by incorporating training data from different unseen classes over time. Existing methodologies have concentrated mainly on employing exemplar-free [25, 28, 32] and memory-based [15, 34–36, 62] solutions to mitigate catastrophic forgetting. Specifically, the former one introduces additional regularization terms to the loss function. While the later one leverages external memory modules or knowledge distillation to selectively replay past data samples, thereby achieving superior results.

As a viable alternative for exemplar-free CIL, Analytic Learning (AL)-based [68–71] methods address the issue of catastrophic forgetting by identifying the iterative mechanism as its main cause and substituting it with linear recursive tools. These methods achieve results comparable to those of memory-based techniques and exhibit robust performance. Nonetheless, the current AL-based methods may encounter the challenge of under-fitting due to their reliance on a single linear projection and the frozen backbone.

While prior Class-incremental learning (CIL) works are mostly limited to a single modality and cannot acquire concise multi-modal representations with the awareness of different classes, there is a growing need to extend these methods to broader domains and modalities. To this end, a few attempts have been made in Audio-visual Class-incremental learning (AVCIL), especially in the field of audio-visual event classification [40, 46]. Specifically, they utilize class token distillation and continual grouping to prevent knowledge forgetting of previous tasks, thereby improving the model's adaptability to capture new and discriminative audio-visual event categories. However, these works explore the data from previous tasks, making them not feasible in privacy-sensitive scenarios with confidential data (e.g., in private property and restricted area), as illustrated in Figure 1.

To solve the aforementioned issues, we propose a novel yet practical framework, namely Multi-Modal Analytic Learning (MMAL), for audio-visual CIL. Specifically, MMAL contains one modality fusion module re-formulating the AVCIL problem in a Recursive Least-Square (RLS) manner, and one Modality-Specific Knowledge Compensation (MSKC) module remedying the under-fitting limitation of the fusion module due to the frozen audio-visual backbone and linearity (i.e., the inherent properties of analytic learning). This study advances AL-based CIL methods by introducing multi-modal incremental learning for the first time and proposes a compensation module that simultaneously leverages individual modality knowledge (i.e., A-MSKC and V-MSKC), thereby addressing their inherent limitations without sacrificing the fundamental advantages. Our key contributions can be summarized as follows.

(1) This paper is a pioneering work to provide the technical route of analytic learning to address the challenging multimodal exemplar-free AVCIL problem.
(2) The fusion module of MMAL re-formulates the AVCIL problem into a RLS problem by freezing the audio-visual backbone. This allows the module to emulate its joint training counterpart that adopts data from both current and historical incremental steps.
(3) To address the intrinsic under-fitting issue associated with analytic learning, we propose the Modality-Specific Knowledge Compensation (MSKC) module. This allows for the simultaneous integration of distinct audio and visual modality information for compensation, enhancing the model's overall learning capability.
(4) Through extensive experiments on three widely-used benchmark audio-visual datasets, AVE, Kinetics-Sounds, and VG-GSound100, we demonstrate the superiority of our MMAL over state-of-the-art competitors in AVCIL scenarios.

## 2 RELATED WORK

### 2.1 Audio-Visual Learning

Previous research has extensively explored the field of audio-visual learning, with numerous methods [3, 5, 30, 38, 42, 50] aiming to explore their temporal correlations, enhancing and complementary characteristics in synchronized streams. A comprehensive review has been presented [57], which indicates the key point in audio-visual learning is establishing cross-modal alignment by repelling non-matching audio-visual embeddings while attracting the matching ones. This draws increasing insight into interdisciplinary fields such as speech separation [17–19, 43, 63], active speaker detection [10, 29, 33, 53] and sound localization [41, 47, 48, 51, 58]. In this paper, our primary objective is to develop compact audio-visual representations with non-stationary audio-visual pairs, specifically for CIL tasks. This task poses greater challenges compared to the aforementioned tasks because of the complexities arising from sequential tasks and feature adaptation to changing audio-visual inputs over time. There are a few attempts [40, 46] on AVCIL addressing above challenges, however, they still require to store historical samples.

### 2.2 Class-Incremental Learning

The primary objective of CIL is to develop strategies that mitigate the negative impact of forgetting previously learned classes while accommodating the introduction of new classes. Existing CIL methods can be categorized into two main types, i.e., memory-based, and exemplar-free methods.

**Memory-based Work**. Memory-based methods [6, 9, 49, 64] leverage additional data, such as exemplars/memory, to tackle CIL. In particular, they leverage the stored data from previous tasks to reinforce learning and prevent catastrophic forgetting. The mechanism was first introduced by iCaRL [49], followed by various attempts due to its superior performance. In [59], an additional trainable layer was designed to correct the bias towards new classes. The LUCIR [21] creates an innovative adaptation which replaces the softmax layer with a cosine layer. In PODNet [15], a spatial-based distillation loss is introduced to retain previously acquired knowledge while accommodating new information. While the above memory-based methods have obtained satisfactory results, they still have the requirement of storing previous samples.

**Exemplar-free Work**. Exemplar-free methods avoid revisiting historical samples during training, which can be mainly categorized into regularization-, and prototype-, and the recent analytic learning-based types, which are discussed as follows.

*1) Regularization-based methods* introduce additional constraints to the learning process to minimize the impact of new tasks on previously learned knowledge. For example, EWC [28] protects the critical parameters associated with previous tasks, ensuring their stability and minimizing their susceptibility to change during training for new tasks. Based on EWC, RWalk [11] defines a Riemannian metric to calculate a penalty term for the loss function, thereby restricting the magnitude of changes in the parameters. Other examples include LfL [25] which penalizes differences in network activations, and LwF [32] prevents activation changes between old and new networks.

*2) Prototype-based methods* use past class prototypes to prevent forgetting. For example, prototype augmentation is proposed in PASS [66] to improve the discrimination of the classes learned in different incremental steps. A prototype selection mechanism is introduced in SSRE [67]. FeTrIL [45] combines a fixed feature extractor and a pseudo-feature generator to improve the stability-plasticity balance.

*3) Analytic learning-based methods* draw inspiration from [20, 69] which leverage least squares to obtain closed-form solutions for network training. In particular, ACIL[71] first transforms CIL into

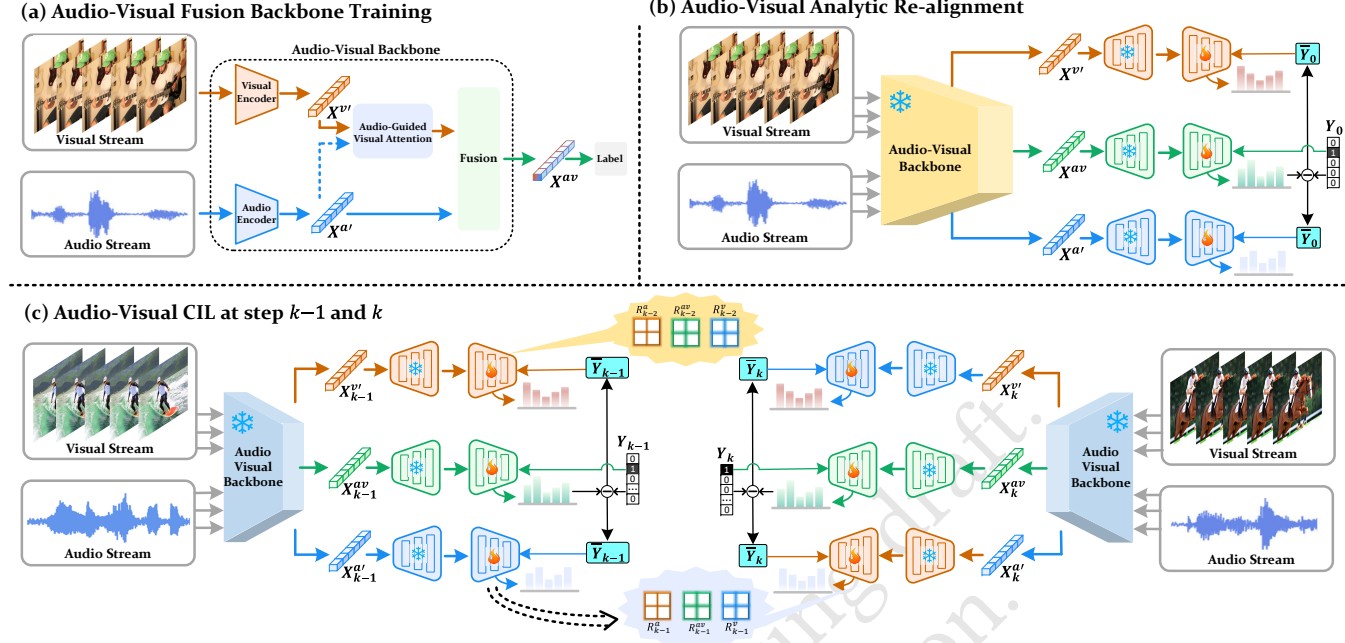

Figure 2: Overall framework of our proposed MMAL, starting with (a) audio-visual fusion backbone training via BP on the base classes, followed by (b) audio-visual analytic re-alignment, and finally (c) audio-visual CIL steps. At each incremental step, it consists of a fusion module (green part) for AVCIL in a RLS manner, and a MSKC module (red and blue parts) to remedy the under-fitting limitation in the fusion module.

a recursive analytic learning process which releases the need of storing exemplars by preserving a correlation matrix. DS-AL [68] employs a dual-stream approach to further enhance the fitting ability of ACIL. The following work GKEAL [70] specializes in the few-shot CIL setting by introducing a Gaussian kernel process.

As an emerging and promising CIL branch, AL-based methods demonstrate impressive performance, even on few-shot scenarios. However, existing AL-based CIL works primarily focus on exploring single-modality solutions, neglecting the potential benefits of multi-modal learning, particularly in audio-visual CIL scenarios. Thus, the integration of audio and visual modalities remains largely unexplored. Furthermore, the AL-based methods require freezing the backbone during the incremental steps, which will cause serious under-fitting limitation due to the complexity of audio-visual data. As a result, we are motivated to explore the multi-modal analytical CIL solution in audio-visual settings.

## 3 PROPOSED METHOD

### 3.1 Preliminaries

Class-incremental learning aims to train the model $\mathcal{F}_\Theta$ with parameters $\Theta$ through a sequence of $K$ tasks $\{\mathcal{T}_1, \mathcal{T}_2, \ldots, \mathcal{T}_K\}$. In AVCIL setting, for an incremental step $\mathcal{T}_k$, its corresponding training set can be denoted as $\mathcal{D}_k^{\text{train}} \sim \{X_{k,a}^{\text{train}}, X_{k,v}^{\text{train}}, Y_k^{\text{train}}\}$, where $X_{k,a}$ and $X_{k,v}$ are the sample's audio and visual modalities respectively in $\mathcal{D}_k$, and $Y_k \in C_k$ is the corresponding label, where $C_k$ is the label space of task $\mathcal{T}_k$. The number of samples of $\mathcal{D}^{\text{train}}$ is $N_k$. For any two tasks' training label space, $C_{k1}$ and $C_{k2}$, they are mutually

exclusive, i.e., $C_{k1} \cap C_{k2} = \varnothing$. The objective of AVCIL at incremental step $k$ is to train the networks given $\mathcal{D}_k^{\text{train}}$, and test them on $\mathcal{D}_{0:k}^{\text{test}}$ (with $\mathcal{D}_k^{\text{test}} \sim \{X_{k,a}^{\text{test}}, X_{k,v}^{\text{test}}, Y_k^{\text{test}}\}$) consisting of data from all seen classes up to step $k$. More importantly, for exemplar-free CIL, exemplar/memory from previous tasks are not allowed to be used during the training of task $\mathcal{T}_k$.

### 3.2 Analytic Audio-Visual Incremental Learning

In this subsection, we will give details of the fusion module of our proposed method. The fusion module aims to transform the traditional AVCIL task into a RLS task and give its analytic solution in a one-epoch training style. It starts with an audio-visual fusion backbone via conventional backpropagation (BP) iterative training, followed by the audio-visual analytic re-alignment, and finally the audio-visual CIL procedures, as illustrated in Figure 2.

**Audio-Visual Fusion Backbone Training via BP.** Given an input sequence of visual frames $\mathbf{x}^v$ and the corresponding audio signal $\mathbf{x}^a$, we first use the pre-trained audio and visual encoder (e.g., AudioMAE [24] and VideoMAE [56]) to extract high-level visual and audio embeddings, respectively. Afterward, the audio-guided visual attention mechanism is employed to aggregate the visual embedding by considering both spatial and temporal dimensions. This has been demonstrated to be highly effective in capturing correlations between audio and visual features [54]. Lastly, we add the audio-guided visual embedding and the audio embedding as the joint audio-visual embeddings and then feed into the classifier. The model is trained on the base classes using backpropagation (BP)

for multiple epochs. Subsequently, when provided with an input sample, the network produces the following output:

$$Y = f_{\text{softmax}}(\mathcal{F}_\Theta(X_a, X_v)W_{\text{FFN}}), \tag{1}$$

where $\mathcal{F}_\Theta(X_a, X_v)$ denotes the output of the audio-visual backbone, $W_{\text{FFN}}$ are the parameters of the linear feed-forward classifier, and $f_{\text{softmax}}$ denotes the softmax function.

Once the audio-visual fusion backbone is trained on the base classes, we freeze this backbone and replace the final linear classifier with a 2-layer analytic network for re-alignment and the forthcoming incremental steps, as detailed in the following section.

**Audio-Visual Analytic Re-alignment**. With the network trained on the base classes, MMAL seeks to detach the backbone and attach it with a 2-layer linear feed-forward network, in which the first layer conducts embedding up-sampling before feeding to the subsequent layer for classification. This procedure enables the network's learning to match the learning dynamics of analytic learning.

Specifically, we feed the input samples $(X_{0,a}^{\text{train}}, X_{0,v}^{\text{train}})$ of the base classes into the backbone to extract the joint audio-visual embeddings, and then up-sample the embeddings followed by an activation function:

$$X_0^{av} = f_\sigma(\mathcal{F}_\Theta(X_{0,a}^{\text{train}}, X_{0,v}^{\text{train}})W_{up}^{av}), \tag{2}$$

where $W_{up}^{av}$ denotes the parameters of the up-sampling layer, $X^{av} \in \mathbb{R}^{N_0 \times d_{av}}$ with $d_{av}$ being the up-sampling size, and $f_\sigma$ is the activation function. Here, we simply follow other analytic methods [69, 71] to adopt the random projection for $W_{up}^{av}$ and *ReLU* for $f_\sigma$.

We then map the up-sampled embeddings into the label matrix $Y_0^{\text{train}}$ via the linear classifier layer, whose weights can be computed by solving:

$$\underset{W_0^{av}}{\text{argmin}} = \| Y_0^{\text{train}} - X_0^{av} W_0^{av} \|_F^2 + \eta \| W_0^{av} \|_F^2, \tag{3}$$

where $\| \cdot \|_F$ is the Frobenius form, and $\eta$ is the regularization term. Then, the optimal solution can be obtained as:

$$\hat{W}_0^{av} = ((X_0^{av})^T X_0^{av} + \eta I)^{-1} (X_0^{av})^T Y_0^{\text{train}}, \tag{4}$$

where $\hat{W}_0^{av}$ is the estimated parameters of the linear classifier layer, and $\cdot^T$ indicates the transpose operation.

**Analytic Audio-Visual Class-Incremental Learning**. Following the analytic re-alignment of the joint audio-visual embeddings, we then move to the AVCIL steps in an analytic learning manner.

Specifically, the learning problem using all seen data at step $k-1$ can be extended from (3) to:

$$\underset{W_{k-1}^{av}}{\text{argmin}} = \| Y_{0:k-1}^{\text{train}} - X_{0:k-1}^{av} W_{k-1}^{av} \|_F^2 + \eta \| W_{k-1}^{av} \|_F^2, \tag{5}$$

where

$$Y_{0:k-1}^{\text{train}} = \begin{bmatrix} Y_0^{\text{train}} & 0 & 0 & \cdots & 0 \\ 0 & Y_1^{\text{train}} & 0 & \cdots & 0 \\ & & & \vdots & \\ 0 & 0 & \cdots & 0 & Y_{k-1}^{\text{train}} \end{bmatrix}, X_{0:k-1}^{av} = \begin{bmatrix} X_0^{av} \\ X_1^{av} \\ \vdots \\ X_{k-1}^{av} \end{bmatrix}$$

Similar to (4), the solution to (5) can be obtained as:

$$\hat{W}_{k-1}^{av} = ((X_{0:k-1}^{av})^T X_{0:k-1}^{av} + \eta I)^{-1} (X_{0:k-1}^{av})^T Y_{0:k-1}^{\text{train}}, \tag{6}$$

where $\hat{W}_{k-1}^{av} \in \mathbb{R}^{d_{av} \times \sum_{i=1}^{k-1} d_{y_i}}$.

To goal of AVCIL under exemplar-free constraint is to sequentially learn new tasks on $\mathcal{D}_k^{\text{train}}$ given a network trained on $\mathcal{D}_{0:k-1}^{\text{train}}$. Nevertheless, the equation above shows that previous data is still required. To reduce this dependency, let $R_{k-1}^{av} = ((X_{k-1}^{av})^T X_{k-1}^{av} + \eta I)^{-1}$, we redefine the CIL process as a RLS task as outlined in the subsequent theorem.

**Theorem 1.** Given training data $D_k^{\text{train}}$ and estimated weights of the final classifier layer $\hat{W}_{k-1}^{av}$ of task $\mathcal{T}_{k-1}$, $\hat{W}_k^{av}$ can be recursively obtained by:

$$\hat{W}_k^{av} = \hat{W}_{k-1}^{av} - R_k^{av}(X_k^{av})^T X_k^{av} \hat{W}_{k-1}^{av} + R_k^{av}(X_k^{av})^T Y_k^{\text{train}}, \tag{7}$$

where

$$R_k^{av} = R_{k-1}^{av} - R_{k-1}^{av}(X_k^{av})^T (X_k^{av} R_{k-1}^{av}(X_k^{av})^T + I)^{-1} X_k^{av} R_{k-1}^{av}, \tag{8}$$

*proof.* See the Supplementary Material.

Theorem 1 suggests that the weights of joint training can be obtained by recursively training on the data from $\mathcal{D}_1^{\text{train}}$ to $\mathcal{D}_k^{\text{train}}$ sequentially, which implies that, by freezing the audio-visual backbone, the AVCIL is equalized to its joint training counterpart as shown in the theorem. This means the model trained incrementally yields the same weights as that trained on both current and all previous data.

## 3.3 Modality-Specific Knowledge Compensation

The audio-visual fusion module is built on the joint audio-visual embeddings. Moreover, since the AVCIL problem is reformulated to a RLS task, the audio-visual backbone has to be frozen, except the final linear classifier. However, when the training samples are complex, the fusion module might not be sufficient to capture the complementary representations of the multi-modal data, thus the under-fitting might occur. To alleviate this limitation, we introduce the MSKC module by leveraging the individual information from audio and visual modality to enhance the fusion module.

This MSKC module contains an audio MSKC (A-MSKC) and a visual MSKC (V-MSKC) sub-module, which operate in a similar manner to the main fusion module, except that the label matrix for updating $\hat{W}_k^{av}$ is generated using the residue from the fusion module. For brevity, in the following part, we take the A-MSKC for explanation while the V-MSKC follows the same rules.

**Audio-Visual Analytic Re-labelling**. Without loss of generality, assume that we have conducted the fusion stream at step $k$, (i.e., obtaining $\hat{W}_k^{av}$ and $R_k^{av}$) and the A-MSKC sub-module at step $k-1$ (i.e., obtaining the weights matrix $\hat{W}_{k-1}^a$ and its corresponding $R_{k-1}^a$). Let $\widetilde{Y}_k$ be the residue after conducting the fusion module, i.e.,

$$\widetilde{Y}_k = [0_{N_{0:k-1} \times d_{y_{k-1}}} \ Y_k^{\text{train}}] - X_k^{av} \hat{W}_k^{av}, \tag{9}$$

where the zero matrix is due to the mutually exclusive AVCIL setting. Let $X_k^a$ be the output the corresponding up-sampled audio embedding, i.e.,

$$X_k^a = f_\sigma(X_k^{a'} W_{up}^a), \tag{10}$$

where $W_{up}^a$ represents the weights of the up-sampling layer of the A-MSKC sub-module.

The $\widetilde{Y}_k$ can be viewed as the residual error, where the joint audio-visual embedding cannot reach. The key idea of the MSKC module is to leverage the individual information from both audio and visual modality to remedy this error, attempting to further reduce the

---

**Algorithm 1** The procedure of MMAL

---

**Require:** Training data $\mathcal{D}_{0:K}^{\text{train}}$, regularization weight $\eta$, audio compensation weight $\lambda_a$, and visual compensation weight $\lambda_v$.

1: **Audio-Visual Fusion Backbone Training**: Train the audio-visual fusion backbone via BP on the base classes.

2: **Audio-Visual Analytic Re-alignment**: Obtain the base weight $\hat{W}_0^{av}$ using (4) and $R_0^{av} = ((X_0^{av})^T X_0^{av} + \eta I)^{-1}$, $R_0^a = ((X_0^a)^T X_0^a + \eta I)^{-1}$, $R_0^v = ((X_0^v)^T X_0^v + \eta I)^{-1}$.

3: **Audio-Visual Analytic Re-labelling**: Obtain the new labels for MSKC module using (11).

4: **for** k=1 to K (with $\mathcal{D}_k^{\text{train}}$, $\hat{W}_{k-1}^{av}$, $\hat{W}_{k-1}^a$, $\hat{W}_{k-1}^v$, $R_{k-1}^{av}$, $R_{k-1}^a$, and $R_{k-1}^v$) **do**

5:     i) Update $R_k^{av}$, $R_k^a$, and $R_k^v$, using (8), (13) and (15) respectively.

6:     ii) Update $\hat{W}_k^{av}$, $\hat{W}_k^a$, and $\hat{W}_k^v$, using (7), (12), and (14) respectively.

7: **end for**

---

forgetting and improve the incremental accuracy. To achieve this, we construct an audio recursive CIL and a visual recursive CIL procedures for audio and video, respectively. Akin to the audio-visual fusion module indicated in Theorem 1, the A-MSKC module follows a similar recursive procedure.

Before conducting the following step, note that $\widetilde{Y}_k$ is the label residual error containing both the current and all previous data, however, under the exemplar-free constraint, we only have access to the data $\mathcal{D}_k^{\text{train}}$ at step $k$. Therefore, to prevent the false supervision for the data at step $k$, let

$$\overline{Y}_k = [0_{N_{0:k-1} \times d_{y_{k-1}}} \quad \{\widetilde{Y}_k\}_{d_{y_k}}], \tag{11}$$

only contain the last $d_{y_k}$ columns for present data $\mathcal{D}_k^{\text{train}}$.

**Audio MSKC**. Upon obtaining the input $X_k^a$ and new labels $\overline{Y}_k$, we can proceed to recursively update the weight $W_k^a$ of A-MSKC following the same procedure indicated in Theorem 1. Then we have

$$\hat{W}_k^a = \hat{W}_{k-1}^a - R_k^a (X_k^a)^T X_k^a \hat{W}_{k-1}^a + R_k^a (X_k^a)^T \overline{Y}_k, \tag{12}$$

where

$$R_k^a = R_{k-1}^a - R_{k-1}^a (X_k^a)^T (X_k^a R_{k-1}^a (X_k^a)^T + I)^{-1} X_k^a R_{k-1}^a, \tag{13}$$

**Visual MSKC**. Similarly, to leverage visual modality to compensate the fusion module, we have

$$\hat{W}_k^v = \hat{W}_{k-1}^v - R_k^v (X_k^v)^T X_k^v \hat{W}_{k-1}^v + R_k^v (X_k^v)^T \overline{Y}_k, \tag{14}$$

where

$$R_k^v = R_{k-1}^v - R_{k-1}^v (X_k^v)^T (X_k^v R_{k-1}^v (X_k^v)^T + I)^{-1} X_k^v R_{k-1}^v, \tag{15}$$

Finally, during the inference stage, the predictions can be obtained as follows:

$$\hat{Y}_k = X_k^{av} \hat{W}_k^{av} + \lambda_a X_k^a \hat{W}_k^a + \lambda_v X_k^v \hat{W}_k^v, \tag{16}$$

where $\lambda_a$ and $\lambda_v$ denote the compensation ratio of audio and visual modality respectively, indicating the degree to which the network relies on the audio and video to compensate and make more accurate prediction. The MMAL is summarized in Algorithm 1.

## 4 EXPERIMENTS

### 4.1 Datasets

We conduct experiments with our proposed method compared to state-of-the-art baselines on three public datasets i.e., AVE [54], Kinetics-Sounds [4] and VGGSound [13]. Specifically, the AVE dataset consists of 4K 10-seconds videos from 28 audio-visual event classes. The Kinetics-Sounds dataset contains around 24K 10-seconds videos from 31 human action classes, while the VGGSound dataset contains around 200K 10-seconds YouTube videos from 309 classes. We follow the experimental protocol adopted in [46], in which 30 classes are randomly selected from the Kinetics-Sounds containing 23K samples in total, and 100 classes (i.e., VGGSound100) are randomly selected from the original VGGSound dataset containing 60K samples in total. For each class of the VGGSound100, 50 samples are randomly selected for validation and testing, respectively.

In terms of class-incremental setting, since the AVE and Kinetics-Sounds datasets contain few classes, we evenly divide the AVE into 4 incremental steps, each of which contains 7 classes, and divide the Kinetics-Sounds into 5 incremental steps, each of which contains 6 classes. While for the VGGSound100 dataset, we explore two different incremental scenarios: 0Base-10Task, and 50Base-$n$Task. The former evenly splits 100 classes into 10 tasks, which mainly follows the setting in [46], while the latter first selects 50 classes as the base classes and then distributes the other across $n$ tasks. It shoud be noted that for the AVE, Kinetics-Sounds and the 0Base-10Task setting on VGGSound100, the base classes are considered to be those classes of the first task. Most existing methods only report small-phase results, e.g., those of $K = 4, 5, 10$, we include $K = 25, 50$ as well to validate MMAL's large-step performance.

### 4.2 Evaluation Metric

The key to successful incremental learning lies in maintaining a delicate balance between plasticity and stability, enabling the acquisition of new knowledge without forgetting previously learned information. Therefore, two metrics, namely average incremental accuracy ($Acc$) and performance drop rate ($PD$), are used to evaluate CIL methods. $Acc = \frac{1}{K+1} \sum_{k=0}^{K} Acc_k$, where $Acc_k$ indicates the average test accuracy of the model incrementally trained at step $k$ by testing it on all seen classes (i.e., $\mathcal{D}_{0:k}^{\text{test}}$), evaluates the overall performance of CIL algorithms. A higher $Acc$ score is preferred. $PD = Acc_0^0 - Acc_K^0$, where $Acc_K^0$ denotes the average accuracy at last step $K$ by testing it on the base classes $\mathcal{D}_0^{\text{test}}$, reveals the degree to which a CIL method forgets the base classes in the first step, which can reflect the degree of the model's retention ability of the old knowledge.

### 4.3 Implementation Details

We conduct all our experiments with PyTorch [44]. For the audio encoder and visual encoder, we use the recent self-supervised pre-trained AudioMAE [24] and VideoMAE [56], respectively. Following the protocol of AudioMAE, the raw audio waveform is transformed into 128-dimensional spectrogram with a 25ms Hanning window and a 10ms shift before feeding into the audio encoder. Similar to VideoMAE, 16 frames are randomly selected from the

Table 1: The overall results of different incremental approaches on the AVE, Kinetics-Sounds, and VGGSound100 datasets. The evaluation metrics are *Acc* and *PD*. The bold part denotes the overall best results, and the underlined part denotes the best results of the compared baselines. The experimental results show that our MMAL achieves the SOTA incremental performance over other methods on all three datasets.

| Method | Exemplar-free? | AVE | | Kinetics-Sounds | | VGGSound100 | | | |
|---|---|---|---|---|---|---|---|---|---|
| | | 0Base-4Task | | 0Base-5Task | | 0Base-10Task | | 50Base-10Task | |
| | | *Acc* (%) ↑ | *PD* (%) ↓ | *Acc* (%) ↑ | *PD* (%) ↓ | *Acc* (%) ↑ | *PD* (%) ↓ | *Acc* (%) ↑ | *PD* (%) ↓ |
| Fine-tuning | ✓ | 42.40 | 76.81 | 41.18 | 90.01 | 26.21 | 81.40 | 53.02 | 59.52 |
| LwF [32] | ✓ | 58.07 | 49.04 | 65.54 | 40.05 | 59.34 | 33.00 | 55.66 | 11.64 |
| ACIL [71] | ✓ | 70.56 | 21.16 | 72.64 | 23.86 | 73.15 | 19.01 | 71.87 | 11.28 |
| iCaRL-NME [49] | ✗ | 56.15 | 43.27 | 64.51 | 44.08 | 56.19 | 40.99 | 55.28 | 29.96 |
| iCaRL-FC [49] | ✗ | 65.88 | 37.50 | 65.54 | 44.62 | 64.22 | 42.40 | 60.25 | 33.28 |
| SS-IL [2] | ✗ | 61.94 | 33.66 | 69.71 | 25.00 | 69.20 | 28.00 | 66.75 | 16.60 |
| AFC-NME [26] | ✗ | 68.46 | 52.89 | 69.13 | 54.03 | 61.41 | 63.60 | 58.00 | 20.68 |
| AFC-LSC [26] | ✗ | 65.21 | 60.58 | 67.02 | 54.57 | 57.76 | 58.20 | 56.22 | 28.88 |
| AV-CIL [46] | ✗ | 74.04 | 22.12 | 73.06 | 22.04 | 72.80 | 21.80 | 67.83 | 14.63 |
| MMAL | ✓ | **76.71** | **18.82** | **78.98** | **18.72** | **76.19** | **17.20** | **74.19** | **8.64** |

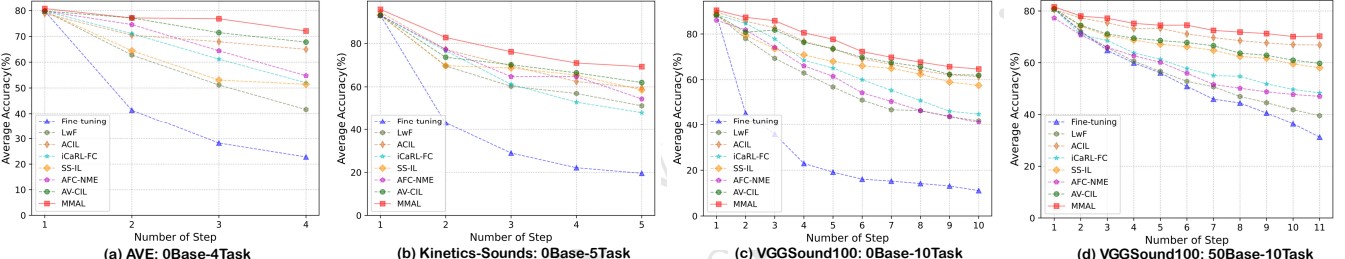

(a) AVE: 0Base-4Task (b) Kinetics-Sounds: 0Base-5Task (c) VGGSound100: 0Base-10Task (d) VGGSound100: 50Base-10Task

Figure 3: Testing accuracy at each incremental step on (a) AVE, (b) Kinetics-Sounds, and (c),(d) VGGSound100 (i.e., 0Base-10Task and 50Base-10Task). The results show that as the incremental step increases, our MMAL generally outperforms other state-of-the-art incremental learning methods.

video clip and then feed into the visual encoder. During the conventional BP training on the base classes, we freeze the pre-trained audio and visual encoder, and only train the remaining parts containing audio-guided visual attention layer, the fusion layer and the final classifier, with a maximum training epochs of 200. We use Adam [27] to optimize the model with learning rate and weight decay of 1e-3 and 1e-4, respectively, with a batch size of 256. All experiments are conducted using one Nvidia A100 GPU with the results averaged over 3 runs. Note that in our MMAL, the audio-visual backbone is only trained during the BP-based training. After which, the parameters of the backbone are fixed and used as a feature extractor during the following incremental steps.

For the regularization parameter $\eta$, we fix it at $\eta = 1$ for all three datasets. For the embedding up-sampling dimension of the fusion module, audio MSKC and visual MSKC (i.e., $d_{av}, d_a$ and $d_v$), we set to (8K, 15K, 15K), (8K, 15K, 15K), and (20K, 20K, 20K) for AVE, Kinetics-Sounds and VGGSound100, respectively. For the compensation ratio of audio and visual modality ($\lambda_a, \lambda_v$) in (16), we set to (0.8, 0.9), (0.5, 1.0), (0.7, 1.0) for AVE, Kinetics-Sounds and VGGSound100, respectively.

## 4.4 Main Results

To demonstrate the effectiveness of the proposed MMAL, we comprehensively compare it to previous representative and state-of-the-art baselines: 1) Fine-tuning: the simplest incremental learning method which initializes the model with the parameters trained from the last step and re-train it on the current step without any constraints to prevent the catastrophic forgetting issue. 2) LwF [32], an exemplar-free method that preserves outputs of previous examples to reduce the forgetting of the old task and act as a regularizer for the new task. 3) ACIL [71]: an exemplar-free AL-based method that identifies the iterative mechanism as the primary cause of catastrophic forgetting and replace it with linear recursive tools. 4) iCaRL [49]: an memory-based method that uses the exemplars in combination with distillation to avoid forgetting. We report the experimental results with both nearest-mean-of-exemplars (NME) classification strategy and the classifier, denoted as iCaRL-NME and iCaRL-FC, respectively. 5) SS-IL [2]: an exemplar-based method that consists of separated softmax output layer combined with task-wise knowledge distillation network. 6) AFC [26]: a knowledge distillation method that minimizes the upper bound of the expected loss increased over the previous tasks. We also report

Table 2: The evolution of average accuracy $Acc$ and last step accuracy $Acc_K$ with the growing incremental step on the VGGSound100 dataset.

| Num. Task | $Acc$ (%) ↑ | $Acc_K$ (%) ↑ |
|---|---|---|
| 5 | 74.57 | 70.30 |
| 10 | 74.19 | 70.28 |
| 25 | 73.95 | 70.32 |
| 50 | 73.96 | 70.31 |

Table 3: Accuracy comparison of our MMAL w/ and w/o the MSKC module.

| Module | AVE | | Kinetics-Sounds | | VGGSound100 | |
|---|---|---|---|---|---|---|
| | $\mathcal{D}_0^{test}$ | $\mathcal{D}_{1:K}^{test}$ | $\mathcal{D}_0^{test}$ | $\mathcal{D}_{1:K}^{test}$ | $\mathcal{D}_0^{test}$ | $\mathcal{D}_{1:K}^{test}$ |
| w/o MSKC | 58.65 | 67.59 | 78.49 | 54.92 | 74.60 | 59.87 |
| w/ MSKC | 57.69 | 76.90 | 76.88 | 67.53 | 73.20 | 63.71 |

the experimental results with both NME classification strategy and the classifier, denoted as AFC-NME and AFC-LSC, respectively. 7) AV-CIL [46]: an audio-visual incremental learning method that incorporates the dual-audio-visual similarity constraint and visual attention distillation.

We compare the proposed MMAL with the above methods in Table 1. We can see that our proposed MMAL outperforms recent state-of-the-art methods significantly, including both memory-based and exemplar-free methods. Specifically, on the AVE dataset, our MMAL outperforms the state-of-the-art $Acc$ and $PD$ results by 2.67 and 3.30, respectively. For the Kinetics-Sounds dataset, our method outperforms the state-of-the-art method, i.e., AV-CIL, by 5.92 and 3.32 for $Acc$ and $PD$, respectively. For the 0Base-10Task setting on the VGGSound100 dataset, our method has the improvement of 2.32 and 1.81 for $Acc$ and $PD$ over the ACIL. While for the 50Base-10Task scenario, our method yields 2.32 and 2.64 improvement for $Acc$ and $PD$ compared the strong baseline ACIL. These experimental results demonstrate the effectiveness of our proposed method in AVCIL.

Furthermore, we show the testing accuracy at each incremental step of our MMAL and other baselines in Figure 3. It can be observed that our method achieves the best performance at each incremental step on three datasets, showing less forgetting and better accuracy. Moreover, comparing two different settings on the VGGSound100 dataset, the accuracy at last step of the 50Base-10Task achieves 70.28%, even though the accuracy on the base classes is only 81.60% which is much lower than that of the 0Base-10Task (i.e., 90.40%), indicating less forgetting. This might be due to that, in the 50Base-10Task setting, we have more data to train the backbone during the base training, thus encouraging to extract more meaningful representations for the following incremental steps. Note that the backbone is freeze once the base training is finished in our MMAL. In summary, our method has a significant superiority for exemplar-free AVCIL compared to others, which demonstrates the effectiveness of our proposal.

## 4.5 Large-step Performance

In Section 3, we have demonstrated that the MMAL exhibits a step-invariant property. To empirically validate this claim, we conduct

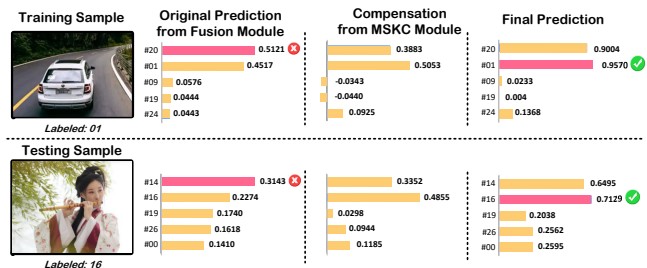

Figure 4: The change of top-5 predictions w/ (left column) and w/o (middle column) the compensation of MSKC module.

large-phase incremental experiments with $K$ from 5 to 50 on the VGGSound100 dataset. The network is first trained on the base classes containing half (i.e., 50) of the full classes, subsequently, the network gradually learns the remaining classes evenly for $K$ steps. For example, for the 10-task experiment, there are five classes per incremental step. As shown in Table 2, the average accuracy $Acc$ and the accuracy at last phase $Acc_K$ almost remain unchanged across various incremental scenarios even under the extreme case of $K = 50$, in which only one class in each incremental step.

## 4.6 Analysis on the MSKC module

The MSKC module utilizes the individual information from audio and visual simultaneously to compensate the limitation of the fusion module. To help understand this strategy, we give specific examples (e.g., 0Base-4Task experiments on the AVE dataset) during the MMAL training and testing. We plot the top-5 predictions with before and after the MSKC module's contribution. As shown in Figure 4, the original prediction for the sample class is inaccurate in the fusion stream. While the MSKC module provides an extra gain, thereby correcting the predicted result. In this example, we can observe a significant prediction change, suggesting a non-trivial enhancement on the MMAL's ability by leveraging the individual information from audio and visual modality.

We also quantitatively analyze the MSKC's impact on the stability and plasticity of the model. To show this, after training on all steps, we evaluate the corresponding model on the base classes $\mathcal{D}_0^{test}$ and the incremental classes $\mathcal{D}_{1:K}^{test}$ separately. As shown in Table 3, by introducing the MSKC module, new classes learned during the incremental steps receive a significant improvement (plasticity) with few performance loss on the base classes (stability). For instance, the MKSC module improves the newly learned classes by 9.31% while only losing the accuracy 0.96% on the base classes of the AVE dataset, suggesting a more reasonable stability-plasticity balance for overall improvement.

## 4.7 Ablation Study

*4.7.1 MSKC Ablation.* To quantitatively evaluate the effect of the MSKC module, we conduct ablation experiments on three datasets. The MSKC contains an audio sub-module and a visual sub-module, aiming to leverage information from audio and video modality, respectively. As shown in Table 4, without the audio compensation (i.e., A-MSKC), the performance will drop 2.50%, 0.99%, and 1.99% on the AVE, Kinetics-Sounds and VGGSound100 respectively. If we

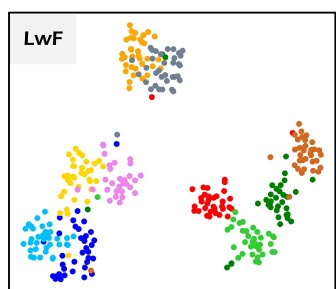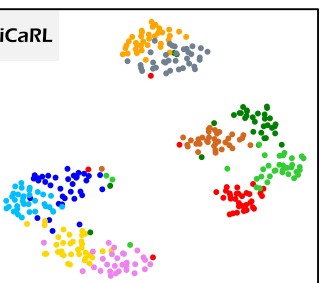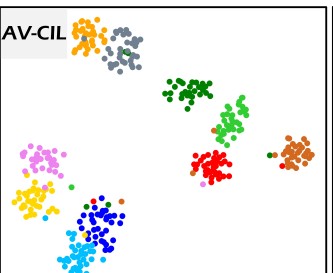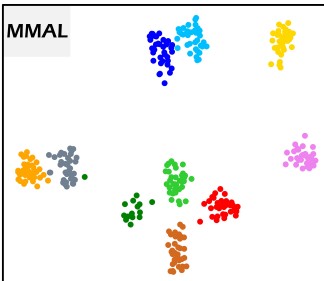

**Figure 5: Qualitative comparisons of representations learned by LwF, iCaRL, AV-CIL and the proposed MMAL. Note that each spot denotes features extracted from one sample, and each color refers to one audio-visual category.**

**Table 4: Ablation study of the MSKC module, containing the A-MSKC and V-MSKC sub-module.**

| Module | AVE | Kinetics-Sounds | VGGSound100 |
|--------|-----|-----------------|-------------|
| MMAL | 76.71 | 78.98 | 76.19 |
| -A-MSKC | 74.21 | 77.99 | 74.20 |
| -V-MSKC | 70.56 | 72.56 | 73.16 |

further remove the visual compensation (i.e., V-MSKC), the average accuracy will yield a significant drop of 6.15%, 6.42% and 3.03% compared to the full model. This demonstrates the effectiveness of our proposed MSKC module.

*4.7.2 Compensation Ratio Ablation.* To explore the appropriate compensation ratio of audio and visual modality to enhance the overall performance, here we conduct ablation experiment to investigate the effect of different ratio on the Kinetics-Sound dataset. As shown in Table 5, when the $\lambda_v$ is set to 1, the best performance is obtained at $\lambda_a = 0.5$. When increasing the ratio of audio modality, the performance starts to decrease, indicating over-compensation could mislead the model. When the $\lambda_a$ is set to 0.5, the performance will drop as the ratio of visual modality becomes small, which suggests that visual modality play an important role to correct the inaccurate predictions of the fusion module.

## 4.8 Qualitative Analysis

Learning informative audio-visual representations with category-aware semantics is critical for classifying audio-visual pairs. To better evaluate the quality of learned category-aware features, we visualize the learned joint audio-visual representations of 9 categories after finishing 4 incremental tasks on Kinetics-Sounds by t-SNE, as shown in Figure 5. It should be noted that each color denotes one class of the audio-visual pair. As can be seen in the last column, audio-visual embeddings extracted by the proposed MMAL are both intra-class compact and inter-class separable. In contrast to our representations in the audio-visual semantic space, mixtures of multiple audio-visual categories still exist among features learned by LwF, iCaRL and AV-CIL. These meaningful visualization results further showcase the superiority of our MMAL in extracting compact audio-visual incremental representations with class-aware semantics for incremental audio-visual learning.

**Table 5: Ablation study of the audio compensation ratio $\lambda_a$ and visual compensation ratio $\lambda_v$ on the Kinetics-Sound dataset.**

| $\lambda_a$ | $\lambda_v$ | $Acc$ (%) ↑ | $Acc_K$ (%) ↑ |
|-------------|-------------|-------------|---------------|
| 0.1 | 1.0 | 78.11 | 67.57 |
| 0.3 | 1.0 | 78.66 | 68.33 |
| 0.5 | 1.0 | 78.98 | 69.31 |
| 0.8 | 1.0 | 78.73 | 69.20 |
| 1.0 | 1.0 | 78.42 | 68.54 |
| 0.5 | 0.8 | 78.48 | 68.84 |
| 0.5 | 0.5 | 77.53 | 66.60 |
| 0.5 | 0.3 | 76.47 | 65.42 |
| 0.5 | 0.1 | 74.69 | 62.97 |

## 5 CONCLUSION

In this paper, we propose a Multi-Modal Analytic Learning (MMAL) to tackle the challenging exemplar-free AVCIL problem for the first time. Specifically, MMAL comprises a fusion module that redefines the AVCIL task through a RLS solution, and a Modality-Specific Knowledge Compensation (MSKC) module that alleviates the underfitting stemming from the static backbone and inherent linearity within the fusion module. Hence, the fusion and compensation module can complement each other for better category-aware semantics during the incremental steps, yielding improved incremental performance. Experimental results on three audio-visual class-incremental datasets AVE, Kinetics-Sounds and VGGSound100 show that our proposed approach outperforms state-of-the-art methods significantly. In the future, we plan to extend our MMAL for tri-modal learning and beyond.

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
