# OpenReview forum: "MMAL: Multi-Modal Analytic Learning for Exemplar-Free Audio-Visual Class Incremental Tasks"
_acmmm.org/ACMMM/2024/Conference — MM2024 Poster_

### Official Review · Reviewer_tmJf · 2024-05-11

**Rating:** 4
**Confidence:** 2

**Summary:**

This paper introduce Multi-Modal Analytic Learning (MMAL) to address the challenging exemplar-free AVCIL for the first time. This method redefines the AVCIL task using a Recursive Least Squares (RLS) and incorporates a Modality-Specific Knowledge Compensation (MSKC) module. This module can mitigate the under-fitting issues caused by the frozen audio-visual backbone and the inherent linearity typical of analytic learning.

**Strengths:**

This paper effectively addresses the challenging exemplar-free AVCIL for the first time. Furthermore, they logically redefine the CIL process as an RLS task through clear mathematical derivation.

**Limitations:**

[1] In Figure 2, I believe additional details may be required for clarity. While the text specifies that there are two layers following the audio-visual backbone, the figure appears to depict multiple layers. Could you please clarify the discrepancy in this representation? Furthermore, in the text (Line 370-371), it is mentioned that the feature $X^{av}$ emerges after upsampling, but this does not seem to be clearly depicted in the figure. Could you please address these discrepancies?

[2] In Line 410-411, it appears that the term $X_{k-1}^{av}$ in the formula for $R_{k-1}^{av}$ should be revised to $X_{0:k-1}^{av}$. Could you please confirm if this modification is necessary?

**Suitability:**

3

---

### Official Review · Reviewer_6eTm · 2024-05-24

**Rating:** 4
**Confidence:** 3

**Summary:**

This work proposed a Multi-Modal Analytic Learning (MMAL) framework for the effective utilization of heterogeneous modalities for the AVCIL task. MMAL re-formulates the AVCIL problem via Recursive Least-Square (RLS) perspective. Then, a Modality-Specific Knowledge Compensation (MSKC) module is to alleviate the under-fitting limitation intrinsic by harnessing individual knowledge from audio and visual modality. Experimental comparisons demonstrate the effectiveness of the proposed MMAL.

**Strengths:**

1. This work explores the novel problem that extends Class-Incremental Learning to Multi-modal Audio-Visual task.
2. Extensive experiments demonstrate the superiority of MMAL over state-of-the-art competitors in AVCIL scenarios.
3. Overall, the paper is well-written.

**Limitations:**

1. Line 361: what is base class? What is the meaning of analytic learning in Audio-Visual Analytic Re-alignment? The object of re-alignment is really confusing.
2. Line 392: "the learning problem using all seen data at step k-1", it seems that Eq. (5) utilizes all data from step 0 to step k-1, which is inconstent with the classical CIL setting, i.e. old data before step k-1 is unseen. Please further clarify it.
3. One of the most important issue is catastrophic forgetting. However, the connection between such issue and the proposed method is weak. How does MMAL address forgetting issue?
4. Please check the Eq. (9), what dose the left term mean?
5. What is the optimization objective of MMAL?
6. The CIL baselines are not up-to-date (LwF and iCaRL)? Any the most recent CIL baselines?

**Suitability:**

3

---

### Official Review · Reviewer_tJL6 · 2024-05-25

**Rating:** 5
**Confidence:** 3

**Summary:**

This paper discusses the challenge of class-incremental learning, particularly in the context of Audio-Visual Class-Incremental Learning (AVCIL). The authors propose a new framework, Multi-Modal Analytic Learning (MMAL), which is an exemplar-free solution that uses a closed-form, linear approach. The MMAL introduces a modality fusion module that redefines the AVCIL problem through a Recursive Least-Square (RLS) perspective. It also includes a Modality-Specific Knowledge Compensation (MSKC) module to alleviate the under-fitting limitation of analytic learning by utilizing individual knowledge from audio and visual modality. The MMAL has shown superior performance in experiments, achieving high accuracy on AVE, Kinetics-Sounds, and VGGSounds100 datasets. Notably, the MMAL provides good data privacy and can better use multi-modal information for improved incremental accuracy.

**Strengths:**

1. MMAL provides the technical route of analytic learning to address the challenging multimodal exemplar-free AVCIL problem.
2. MMAL re-formulates the AVCIL problem into a RLS problem by freezing the audio-visual backbone.
3. MMAL proposes a MSKC module, which allows for the simultaneous integration of distinct audio and visual modality information for compensation, enhancing the model’s overall learning capability.

**Limitations:**

1. The backbone is not frozen in the fusion backbone training phase of the MMAL method, does this have an impact on the generalizability of the whole model? Has it been considered to freeze the backbone itself and only use training with less number of parameters such as linear layers instead of phase (a)? If not, can some explanation be given?
2. Is it possible to add experiments under different backbone?

**Suitability:**

3

---

### Official Review · Reviewer_HWFc · 2024-06-11

**Rating:** 4
**Confidence:** 2

**Summary:**

The authors study the problem of class-incremental learning with an exemplar-free constraint. The authors reformulate audio-visual class incremental learning by recursive least-square and design modlaity-specific knowledge compensation module. The experiments are conducted using AVE, Kinetics-Sounds and VGGSound. They use pre-trained AudioMAE and VideoMAE as audio encoder and visual encoder. The main concern is about the novelty of this paper, I cannot find a lot of novel contribution to audio-visual, compared with visual based ones.

**Strengths:**

1. The problem is interesting.
2. The experimental results are good on AVE, Kinetics-Sounds and VGGSound100.
3. There is some contribution about reformulation of AVCIL problem with RLS, which is interesting.

**Limitations:**

1. The dataset used in this paper is not large enough and it is better for the authors to explore the incremental learning for large audio-visual model.

**Suitability:**

2

---

### Meta-Review · Area_Chair_jJow · 2024-07-01

**Recommendation:** Accept (Poster)
**Confidence:** 3

**Metareview:**

All four reviewers are positive. The proposed method is novel and effective.